# Physical Characterization and Cellular Toxicity Studies of Commercial NiO Nanoparticles

**DOI:** 10.3390/nano12111822

**Published:** 2022-05-26

**Authors:** Filip Kunc, Michael Bushell, Xiaomei Du, Andre Zborowski, Linda J. Johnston, David C. Kennedy

**Affiliations:** 1Metrology, National Research Council Canada, 1200 Montreal Road, Ottawa, ON K1A 0R6, Canada; filipo.kunc@gmail.com (F.K.); mikebushell@cmail.carleton.ca (M.B.); linda.johnston@nrc-cnrc.gc.ca (L.J.J.); 2Energy, Mining and Environment, National Research Council Canada, 1200 Montreal Road, Ottawa, ON K1A 0R6, Canada; xiaomei.du@nrc-cnrc.gc.ca (X.D.); andre.zborowski@nrc-cnrc.gc.ca (A.Z.)

**Keywords:** nickel oxide, toxicity, characterization, size, oxidative stress

## Abstract

Nickel oxide (NiO) nanoparticles from several manufacturers with different reported sizes and surface coatings were characterized prior to assessing their cellular toxicity. The physical characterization of these particles revealed that sizes often varied from those reported by the supplier, and that particles were heavily agglomerated when dispersed in water, resulting in a smaller surface area and larger hydrodynamic diameter upon dispersion. Cytotoxicity testing of these materials showed differences between samples; however, correlation of these differences with the physical properties of the materials was not conclusive. Generally, particles with higher surface area and smaller hydrodynamic diameter were more cytotoxic. While all samples produced an increase in reactive oxygen species (ROS), there was no correlation between the magnitude of the increase in ROS and the difference in cytotoxicity between different materials.

## 1. Introduction

Nickel oxide (NiO) nanoparticles are used in a wide array of industrial applications including in batteries, semiconductors [1,2], textiles [3], solar cells, and other electronic components [4,5]. Human exposure arises from the production of these materials both from the manufacturing processes as well as their use. Exposure may also result from the release of the materials to the environment during end-of-life disposal. Nickel is considered a trace essential element in humans; however, there are no known enzymes in human cells that require nickel [6]. The human requirement is assumed to be exclusively for the maintenance of a healthy microbiota environment. Ni is required for both urease, and hydrogenase, two abundant and critical enzymes for the healthy existence of many micro-organisms [6]. Nickel oxide nanoparticles may also contribute to lung cancer with prolonged exposure via inhalation and have been reported to cause inflammation and oxidative tissue damage [7,8].

The cytotoxicity of NiO nanoparticles has been studied in non-mammalian systems [9,10,11,12], in mammalian cell culture [13,14,15,16,17,18], and in rats [19,20]; however, the cell culture studies typically lack a breadth of materials and therefore cannot identify trends in how particle size or surface coating may be involved in particle cytotoxicity. In order to identify such trends, a range of NiO nanoparticles with varying physical–chemical properties is required, and it is critical to thoroughly characterize the materials using an array of surface and structural techniques. Such an approach allows one to correlate the physical parameters of the nanomaterials to changes in their cytotoxicity in cell culture. This can provide an estimate of how the particles may behave and can identify potential risks posed by particles of specific sizes or compositions. While conflicting results in nanotoxicology can often be attributed to different cell lines or assays, for NiO, even when the same assays are performed using the same cell line [13,14,15], conflicting results arise. This highlights the importance of performing careful and indepth physical characterization, a breadth of biological assays, and using a range of different NiO nanoparticles with different sizes or surface chemistry in order to elucidate trends and determine risk thresholds for these materials. Regulations for the safe use of metal and metal oxide nanoparticles are often based on data for the bulk metal and fail to incorporate the unique properties that the materials may manifest at the nanoscale. Data points on a single species in a single cell line are insufficient to make absolute assessments, and variability in experimental design and reporting makes comparing results from different studies challenging. The lack of robust data sets has made it challenging for regulatory agencies to make informed decisions on the safe use of nanoparticles.

Here, we have purchased eight NiO nanoparticles of different sizes and with different surface coatings from four manufacturers to cover a range of representative materials that might be incorporated into products, and which are readily available to other researchers to improve interlaboratory comparisons of data. Similar nanotoxicology studies often lack a clear understanding of the state of particle dispersion. While deagglomeration of these nanomaterials to single particles may not be feasible in aqueous media, in this study, we optimized the dry powder dispersion in water, which will enhance the reproducibility of the results in future studies. The particle sizes of these dispersions have been measured by dynamic light scattering (DLS) and transmission electron microscopy (TEM). Surface charge, specific surface area, and quantification of surface coatings were assessed by electrophoretic zeta potential measurements, BET (Brunauer-Emmett-Teller) analysis, and NMR, respectively. Using commercial samples as opposed to pristine laboratory synthesized and purified materials is more realistic for applications that use nanomaterials to enhance the performance of consumer products. Nevertheless, this work highlighted the challenges associated with the characterization of heterogeneous nanomaterials and the limitations of current analytical techniques for such non-ideal particle ensembles. These well-characterized dispersions were then used for cytotoxicity studies in two cell lines, and changes in the particle dispersions were monitored in cell culture medium using time-resolved DLS measurements that correlated to the time points used for the cytotoxicity measurements. NiO particles produce significant cytotoxicity which continues to increase over 48 h. These results were not obviously coupled to an increase in reactive oxygen species, suggesting that other modes of action must be important for this cytotoxic effect.

## 2. Experimental Details

### Materials

Five bare NiO nanoparticles were purchased from four different suppliers—Skyspring Nanomaterials Inc. (Houston, TX, USA) (reported size, 50 nm) (Ni-01), mkNano (Mississauga, ON, Canada) (reported size, 90 nm) (Ni-02), Sigma-Aldrich (St. Louis, MO, USA) (<50 nm) (Ni-03), and US Research Nanomaterials (USRN) (Houston, TX, USA) ((reported sizes, 18 nm and 15–35 nm) (Ni-04 and Ni-08, respectively). Three coated NiO particles were also purchased from US Research Nanomaterials. These particles were all reported to be 18 nm in diameter and coated with polyvinylpyrrolidone (PVP) (Ni-05), or stearic acid (Ni-06), or covalently modified with (3-aminopropyl)triethoxysilane (APTES) (Ni-07). Materials from all suppliers except Sigma-Aldrich were received in sealed plastic bags and the powders were transferred into certified particulate free glass jars and stored under ambient conditions.

Deuterium oxide (99.9%), sodium deuteroxide (10 M in D_2_O), methanol-d_4_ (99.9%), DMSO-d_6_ (99.9%), pyridine-d_4_ (99.9%), pentadecafluorooctanoic acid (99.9%), sodium hexametaphosphate (99.9%), TraceCERT maleic acid (99.94% maleic acid mass fraction), and TraceCERT potassium phthalate monobasic (99.92% mass fraction) were purchased from Sigma-Aldrich (Oakville, ON, Canada).

## 3. Physical and Chemical Characterization

### 3.1. Specific Surface Area Measurements

The Brunauer Emmett Teller (BET) method with nitrogen adsorption was used for the measurement of specific surface area (SSA) with an ASAP 2020 system from Micromeritics. The samples were heated at 10 °C/min to 110 °C, held for 10 min, and then heated to 200 °C at 10 °C/min and held for 2 h. Specific surface area was determined by the multipoint BET method.

### 3.2. Dispersion of Nanoparticles

Dry NiO nanoparticles (15 mg) were weighed, and 15 mL of deionized water (Milli Q, 18.2 MΩ cm) was added to give a 1 mg/mL suspension. The sample was vortexed for 5 s and then sonicated to the desired energy using a 130 W ultrasonic processor (EW-04714-50, Cole-Parmer, Quebec, QC, Canada) equipped with a ¼ inch tip probe (EW-04712-14 Cole-Parmer, Quebec, QC, Canada) and operated at 50% amplitude with 30 s on/off cycles. To prevent overheating, the sample was immersed in a water–ice bath during the sonication cycle. The particle size was assessed by DLS as a function of applied sonication energy to determine the optimal sonication energy for dispersion of each material. The delivered sonication energy was calculated using total energy transfer efficiency for the sonicator configuration which was determined calorimetrically to be 0.97. The sonicator probe was polished after transferring a total of 12,000 J to maintain its efficiency. The NiO sample coated with stearic acid was dispersed in ethanol for optimizing the dispersion; however, an aqueous dispersion was prepared and characterized for the cell culture experiments.

### 3.3. Dynamic Light Scattering and Zeta Potential

Sonicated dispersions were transferred to a disposable PMMA DLS cell and analyzed with a Zetasizer Nano ZS (red) (Malvern, Quebec, QC, Canada) using a 632.8 nm HeNe laser and signal detection at 173°. The measurements of Z-average (equivalent hydrodynamic diameter) and polydispersity index (PDI) were performed at 25 °C with automatic positioning and attenuation selection set on. Each sample was equilibrated for 180 s and 3 measurements, each consisting of 10 runs of 10 s, were acquired. Each measurement was analyzed using Zetasizer software (ver. 7.11, Malvern, Quebec, QC, Canada) by the cumulants method with the general purpose model using the values for viscosity and the refractive index of dispersant of *η* = 0.8872 cP and n = 1.330. A range of NiO concentrations (1.00–0.01 mg/mL) was tested; based on the shape of the correlation function, the attenuator, and the count rate values 0.1 mg/mL concentration was deemed optimal for the measurement.

The same instrument was used for the zeta potential measurements. Sonicated dispersions at 0.1 mg/mL in water were placed in a bent capillary DLS cuvette (DTS1061, Malvern, Quebec, QC, Canada). The pH of 0.1 mg/mL dispersions was measured in triplicate using a pH meter.

### 3.4. Transmission Electron Microscopy

Samples were deposited on carbon-film-covered copper grids (200 mesh, Ted Pella 01840-F) that had been plasma-treated (30 sccm 75%/25% Ar/O_2_, 2 min, ≈40 W using a Fischione (1070, NanoClean, Export, PA, USA)**.** An amount of 10 μL of previously dispersed metal oxide dispersion (0.1 mg/mL or 0.01 mg/mL) was added to the carbon film, and wicked away after 10 min using filter paper. The sample was then immersed in deionized water twice and allowed to dry, typically 1–2 h. Alternately, some samples (ethanol suspensions and some aqueous suspensions) were allowed to dry after wicking away the excess metal oxide suspension. For most metal oxides, samples at the two concentrations were tested and the one with the least aggregation was used for imaging. The lower concentration samples did not always give the better dispersed samples for imaging.

TEM images were recorded on a Titan [3] 80–300 FEI microscope (Ottawa, ON, Canada) operated at 300 kV and calibrated with a TEM magnification standard (MAG*I*CAL, EMS). Images were analyzed with ImageJ using the polygon outlining feature to trace individual particles and measure area, perimeter, Feret, and minFeret. Area was converted to equivalent circular diameter and aspect ratio was calculated from the Feret/minFeret ratio.

### 3.5. Surface Coating Quantification by NMR

The procedure for quantitative NMR was followed as described in previous work [21]. The covalently attached functional groups for silane (APTES)-modified metal oxides were removed by hydrolysis of the nanoparticles in 0.4 M NaOD in D_2_O; the samples were dispersed by sonication in an ultrasonic bath, and then placed in a heated shaker at 45 °C/1200 RPM for 24 h. The solution was cooled, the sample was centrifuged to pellet the nanoparticles, and the supernatant was removed and analyzed by NMR after addition of an internal standard (potassium hydrogen phthalate). PVP-coated samples were dispersed in D_2_O containing sodium hexametaphosphate (0.5% by mass) and placed in a heated shaker for 24 h at 45 °C/1200 RPM. The solution was cooled, the sample was centrifuged to pellet the nanoparticles, and the supernatant was removed and analyzed by NMR after addition of an internal standard (potassium hydrogen phthalate). A ligand-exchange approach was used for stearic acid quantification. The weighed powder was dispersed by a short sonication step using an ultrasonic bath in 0.65 mL of methanold-d_4_ and 0.035 mmol of pentadecafluorooctanoic acid. This was added as a DMSO-d_6_ solution (20 µL). The samples were placed in an orbital heated shaker and shaken at 1200 RPM at 45 °C for 24 h. The samples were cooled to room temperature, centrifuged at 14.3 k RPM for 5 min, and the supernatant was separated from the pellet. Prior to the qNMR experiment, the sample was combined with maleic acid internal standard solution in DMSO-d_6_.

### 3.6. Inductively Coupled Plasma Mass Spectroscopy

Samples were analyzed at the University of Ottawa in the Geochemistry Laboratory Facility on an Agilent 8800 triple quadrupole mass spectrometer (Mississauga, ON, Canada).

## 4. Cell Culture Experiments

### 4.1. Cell Culture Preparation

A549 and J774A.1 cells (American Tissue Culture Center, Manassas, VA, USA) were grown in Dulbecco’s modified Eagle’s medium (DMEM, Grand Island, NY, USA) (Gibco, Waltham, MA, USA) supplemented with 10% fetal bovine serum (FBS) (Gibco) and 1% penicillin-streptomycin (Pen/strep) (50 µg/mL, Gibco) under standard culture conditions (37 °C, 5% CO_2_). Cells were grown in T75 flasks (Falcon) and Trypsin-EDTA solution (Gibco) was used for passaging A549 cells (2 mL per T75 flask). For passaging J774A.1 cells, a cell scraper was used to detach cells from flask, and the cells were then diluted one into ten in a new flask.

### 4.2. Nanoparticle Stock Sample and Dilution Preparation

Nanoparticles were suspended in water as reported in the section on physical properties. These same suspensions were then used for the biological experiments. A stock suspension of nanomaterials was prepared by diluting one part aqueous nanoparticle suspension with one part complete DMEM with 10% FBS and 1% pen/strep. This diluted the medium by 50%, so all subsequent dilutions were made with a 1:1 mixture of medium and water so that all experiments would have the same serum concentration in the medium as this will affect the toxicity of both particles and metal ions. The particles were then added 1:1 to wells with cells and medium so the final serum concentration of all experimental wells was always 7.5%.

### 4.3. Dynamic Light Scattering in Cell Culture Medium

Samples were run on a Malvern Zetasizer Nano-ZS in plastic cuvettes (BRAND) with a 1 mL sample volume. Each sample was measured 3 times. Nanoparticles suspended in water and medium (1:1) were mixed with 1 equivalent of medium to achieve the same 7.5% serum concentration that was used in the cell culture experiments, dilutions were then performed using a 75% medium, 25% water solution in order to dilute cells to appropriate concentrations for DLS without adjusting the serum concentration (typically 100 µg/mL). Samples were measured immediately upon preparation and then every 24 h for 3 days. Each day, samples were suspended before measuring as the particles had all settled out of suspension. In some cases, significant settling occurred even during the 10 min needed to run the DLS measurement.

### 4.4. MTT Assay

Cells were seeded into wells in a 96-well plate (Falcon) (1 × 10^5^ cells/mL, 100 µL per well) to cover an 8 × 6 grid, filling 484 wells. Remaining wells were filled with 200 µL of PBS. After 24 h, 100 µL volumes of dilutions of particles in a 50% aqueous dilution of complete medium spanning from 500 µg/mL to 5 µg/mL were added to the seeded wells (final concentrations spanning 250 µg/mL to 2.5 µg/mL). For each nanoparticle, seven dilutions were prepared, and for each dilution, three replicates were performed. In the remaining 6 wells, 100 µL of diluted medium was added as a particle-free control. Cells were then incubated with nanoparticles for 24, 48, and 72 h. For each time point, workup consisted of adding a 50 µL PBS solution of MTT (2.5 mg/mL) to each well and then incubating for 3 h. After 3 h, the medium was aspirated from all wells, leaving purple formazan crystals in those wells with viable cells. To each well, 150 µL of DMSO was added and plates were agitated manually to dissolve the crystals. A total of 100 µL from each well was then transferred to a fresh plate. This was done in order to remove scattering from precipitated particles that affected the readings. Plates were then analyzed using a plate reader (Fluorostar Omega, BMG Labtech., Ortenberg, Germany) to determine the absorbance of each well at 570 nm. This reading divided by the average from the reading of the six control wells was plotted to determine the IC_50_ value of each particle for each cell line. Three replicates were performed for each sample on each cell line at each time point.

### 4.5. Neutral Red Assay

This procedure was modified from a published protocol [22]. Cells were prepared in a manner identical to the MTT assay and treated for 72 h with nanoparticles in a 96-well plate. Neutral red medium was prepared as reported [22]. After 24 h, medium was aspirated, cells were washed once with 150 μL of PBS, and 100 μL of neutral red medium was added to each well. Plates were then incubated for 2 h, at which time the neutral red medium was aspirated from the wells. Cells were then washed with 150 μL of PBS, and then 150 μL of destain solution (50% ethanol, 49% water, 1% acetic acid) was added to each well. Plates were then shaken for 2 min to extract the neutral red from the cells, and then the absorbance of each well was measured at 540 nm.

### 4.6. DCFDA Assay

Cells were prepared in a manner identical to the MTT assay but seeded into black-walled 96-well plates. Immediately prior to use, DCFDA buffer and solution were prepared as per the assay kit protocol. After seeding the cells overnight, the wells were washed with 100 µL DCFDA buffer. The wells were then filled with 100 µL DCFDA solution and incubated in standard culture conditions for 45 min. After incubation, the DCFDA solution was removed and replaced with 100 µL 1X PBS. The fluorescence was read at Ex/Em 485/535 using a spectrophotometer. The buffer was then removed and replaced with 100 µL of cell culture medium. Dilutions were then added in a manner identical to the MTT assay, and plates were scanned immediately to baseline the fluorescence in each well. Fluorescence measurements were taken again at Ex/Em = 485/535 nm after 1, 2, 3, and 4 h. Little effect was observed for all samples, so they were then incubated overnight and the fluorescence was recorded again at 24 h. Between each fluorescence measurement, the plates were incubated under standard culture conditions.

## 5. Data and Results

### 5.1. Physical Characterization

Commercial samples were selected for this study to better gauge the quality of materials available to manufacturers that might incorporate nanomaterials into products and to help build a library of data that could be easily cross referenced with other labs who can purchase the same materials. This type of data can then be used to inform the development of regulations for the safe and effective use of NiO nanoparticles and their incorporation into consumer products. Indeed, funding for this work was provided with the intended purpose to use the data to help guide the development of regulations for the safe use of metal oxide nanoparticles in consumer products in Canada. This study was conducted using eight different materials purchased from four manufacturers, all of which were supplied as dry powders. The samples were selected to represent a range of sizes, suppliers, and surface coatings. These samples included three different surface coatings from a single manufacturer. The nominal particle size and specific surface area supplied by the manufacturers are included in Table 1 along with a summary of our independent determinations for size, surface area, surface charge, and hydrodynamic diameter.

**Dispersibility in water.** The initial dispersion of powdered materials was carried out in deionized water instead of cell culture medium since the generation of reactive oxygen species by sonolysis may lead to the denaturation of proteins [23]. To disperse samples in a manner that can be reproduced in other laboratories, the method established by Deloid et al. was followed [24]. The probe sonicator was calibrated calorimetrically in order to determine the delivered sonication energy. Samples were sonicated with varying energies and plots of Z-average and PDI measured by DLS as a function of sonication energy (Appendix A) were used to determine an appropriate sonication energy. Table 1 summarizes the Z-average and PDI for samples sonicated with the recommended energy. It should be noted that the materials varied significantly in their dispersibility. Some samples, such as Ni-04 (Appendix A), gave the expected decrease of Z-average and PDI with increased sonication at low energy and reached a plateau at higher energies; this made it straightforward to select an appropriate energy that dispersed the sample, but minimized possible damage at high applied energy. By contrast, samples such as Ni-01 and Ni-02 (Appendix A), showed little effect of sonication except for the first one or two points and, otherwise, further sonication lead to DLS measurements with unstable readings that suggested rapid reagglomeration. Both of these samples gave intensity traces that varied from run to run and showed multiple components. Such results (PDI above ~ 0.3), are not suitable for analysis using the conventional cumulants approach and only approximate values for Z-average and PDI are provided in Table 1.

With the exception of Ni-01 and Ni-02, the NiO samples gave Z-averages in the range of 230–380 nm with PDI between 0.27 and 0.37. The large Z-average, compared to the nominal size (Table 1), and high polydispersity indicated that the samples were agglomerated with a wide range of sizes despite attempts to optimize the dispersion procedure. Qualitatively, most samples showed some level of sedimentation after sitting for a period of hours after sonication, consistent with the relatively large sizes. In this regard, these commercial NiO nanoparticles differ from other types of nanomaterials such as gold, silver, or silica nanoparticles which give colloidally stable suspensions with narrow size distributions. It should be noted that DLS is much more sensitive to large particles (intensity proportional to radius [6]) and the agglomerates present in all these samples made it difficult to detect small particles. Examination of the individual intensity histograms provides evidence for a < 100 nm fraction in samples Ni-01, -03, -04, -06, -07, and -08. Therefore, despite the issues with agglomeration, these materials do contain a nanosized fraction that may possess unique nanoscale properties that require toxicological assessment. The discrepancy between the nominal size provided by the supplier and the DLS results also highlights the importance of complimentary TEM characterization to determine the primary particle size.

The surface charge for particles was measured for sonicated particle suspensions in deionized water. The pH of the prepared dispersions was measured since zeta potential measurements are a function of pH (Table 1). Due to its hydrophobic character, the stearic acid-coated sample was dispersed and measured in ethanol; its relatively low surface charge value (13 ± 1 mV) can be explained by stearic acid shielding the surface. It can be noted that all NiO samples have a positive surface charge, which is consistent with the presence of Ni^2+^ ions on the outer layer of the hydrated NiO surface as proposed by Kitakatsu et al. [25]. Interestingly, most samples had a sufficient charge (≈30 mV) to maintain colloidally stable dispersions; however, DLS measurements still indicated sedimentation caused by the presence of large agglomerates.

**Size distribution by TEM.** TEM analysis was employed to obtain information on the size/shape of primary particles for comparison to the manufacturer information and the DLS experiments described above. Samples were dispersed using the recommended sonication energy provided in Table 1 and deposited on TEM grids. Representative images are shown in Figure 1 for each sample, with a second lower magnification image provided in Appendix A. Aggregation was also a significant problem in TEM experiments and varied from sample to sample; the images shown in Figure 1 and Appendix A are for less aggregated sample areas. The level of aggregation observed in the TEM images did not correlate with the level of dispersibility indicated by the DLS measurements. This indicates that further sample agglomeration during the deposition and drying of the sample on the TEM grid is the main factor in determining the level of dispersion observed by TEM. Nevertheless, it was possible to find areas that were suitable for size analysis of individual particles. With the exception of Ni-03 which had a higher aggregation level and lower contrast than the other samples, approximately 100 particles were analyzed for each sample. Box plots showing the distributions for mean equivalent diameter and aspect ratio for each sample are shown in Appendix A. The mean equivalent circular diameter and standard deviation (a measure of the width of the particle size distribution) are provided in Table 1.

All the NiO samples gave a mean equivalent diameter that was less than approximately 20 nm. In most cases (Ni-03–Ni-08), the mean equivalent diameter was close to the nominal value provided by the manufacturer, but it was considerably lower for two samples (Ni-01, Ni-02), Appendix A. A previous study of four families of other commercial metal oxide nanoparticles has also demonstrated that a significant number of samples have primary particle sizes that differ substantially from those reported by the manufacturer [26]. Note that the particle size distributions as assessed by the standard deviation are quite broad for most samples. The broad size distributions and limited statistics suggest that small differences between samples are unlikely to be meaningful. In some cases, the images contained large “features” that were not counted either because they overlapped with other particles, or their edges were ill-defined. In other cases, there were very small, poorly defined particles that were also not counted. Despite these complications, the TEM data confirmed that all samples contained a significant fraction of particles with an equivalent diameter significantly below 100 nm. As illustrated in Figure 1, there are various particles shapes, ranging from approximately circular or elliptical to square or rectangular. However, the aspect ratios are all similar (means of 1.25 to 1.45 with no particles with an aspect ratio greater than ~2), indicating that differences in shape are unlikely to be a consideration for the toxicology studies.

**Specific surface area****(SSA).** SSA values were measured by BET for all samples and compared to the values provided by the manufacturers where available. For most samples, the measured values were similar to those provided by the manufacturer (Table 1). One exception was Ni-06, which had a significantly lower measured SSA (2 ± 1 m^2^/g vs a reported value of 50–100 m^2^/g). An estimated surface area can be calculated from the nominal particle diameter and the material density (6.67 g/cm^3^ provided by US Research Nanomaterials). Calculated values were 18 m^2^/g for 50 nm sample Ni-01, 10 m^2^/g for 90 nm sample Ni-02, 50 m^2^/g for all 18 nm samples, and 26–60 m^2^/g for 15–35 nm sample Ni-08. The SSA measured by BET can indicate the level of aggregation of the dry powders. Low values compared to those estimated from the primary particle size suggest that the particles were strongly aggregated, preventing nitrogen penetration. This can indicate that the material may be difficult to disperse to give a colloidally stable suspension. In contrast, higher values than the estimated surface areas based on the particle diameter may be indicative of increased surface roughness or material porosity.

**Quantification of surface coatings and functional groups.** The selected materials represent three different types of surface coatings: Ni-05 with a physically adsorbed polymer (PVP) (Figure 2), Ni-06 with stearic acid attached as a carboxylate, and Ni-07 covalently modified with a commonly used aminopropylsilane. Although a number of approaches for the quantification of coatings on nanomaterials has been reported, these vary in their range of applicability and sensitivity and validated standard procedures are still not available [27]. We removed the coating from the particles for quantification by quantitative NMR after removal of the particles by centrifugation.

Removal of aminoproylsilane was achieved by basic hydrolysis using the same method optimized previously for a number of commercial ZnO nanoparticles [28]. A ligand exchange process using hexametaphosphate and polyfluorooctanoic acid, respectively, was used to remove the PVP and stearic acid coating. Peaks in the NMR spectra of the material removed from the NiO nanoparticles were readily assigned to the expected coating as shown in Figure 2 for the PVP sample, Ni-05; no additional signals were detected, indicating that the samples do not contain significant levels of organic contaminants. The results of quantification by comparison to an internal standard are provided in Table 2. It should be noted that the accuracy of the qNMR method depends on the efficiency of the functional group hydrolysis or coating desorption. The detailed optimization of the hydrolysis and ligand exchange methods for several metal oxides nanoparticles and a comparison with other quantification methods, including thermogravimetric analysis and x-ray photoelectron spectroscopy, has been reported [29]. This work and earlier studies [21,28] on silica and ZnO nanoparticles have indicated that qNMR has very good sensitivity and reproducibility and can be used to benchmark the reliability of other methods for quantifying surface functional groups and coatings. The previous metal oxide study demonstrated good agreement between the atomic nitrogen content measured by XPS and the qNMR data for APTES-functionalized nanoparticles. However, the presence of adventitious carbon containing contaminants in the commercial materials and the variable hydroxyl content made it impractical to obtain quantitative assessments of stearic acid content using signals in either the C1s or O1s region.

To better illustrate the surface coverage for various materials, the molar content was converted to the number of molecules (for PVP this is the vinylpyrrolidone monomer) per 1 nm^2^. Based on atomic radii of nickel and oxygen (0.125 and 0.071 nm), 1 nm^2^ area is occupied by ≈6.5 Ni atoms. Based on this approximation, the surface of Ni-06 is almost saturated with stearic acid which binds to ≈85% of the Ni atoms (5.5 molecules per nm^2^, Table 2) assuming that all stearic acid is bound at the surface. For comparison, previous XPS studies on single crystal NiO indicated adsorption of one carboxylate for every two nickel sites [30]. It is more difficult to make a similar estimate for the aminopropylsilane functionalized sample. Geometrically, 1 nm^2^ can fit 11 vertically stacked 3-aminopropylsiloxane molecules. However, these have to bind through the surface hydroxyls which are present in variable amounts on the NiO surface. The number of hydroxyls depends on the temperature applied during the synthesis and the humidity postsynthesis and can range between 10 and 85% [25]. The estimated content of 7.6 aminopropyl siloxane moieties per nm^2^ (Table 2) may be higher than the hydroxyl content, in which case some vertical polymerisation of aminopropyl siloxane may occur. For the PVP-coated sample, the molecular weight of the polymer is not known. However, for illustration, 1 nm^2^ can be occupied by ≈5 horizontally oriented ethylenepyrrolidone moieties. Therefore, the estimation of 10.6 vinylpyrolidone molecules per nm^2^ suggests that the surface is completely covered by polymer chains. It should be noted that the NMR studies indicate that PVP rapidly desorbs from the NiO surface in an aqueous environment, which may have implications for studies in cell culture media.

### 5.2. Nanoparticle Purity by ICP-MS

The purity of the nanoparticles was assessed using ICP-MS. The Ni content was assessed accurately using this methodology and metal impurities were identified in three of the particles (Appendix A). From this, it is clear that Ni-01, Ni-02, and Ni-08 all contain measurable impurities that account for at least 0.1% of the total mass of the particles. In the case of Ni-01, there are significant impurities including 19.1% Mn. From this, we can conclude that the behavior of Ni-01 may be significantly different from the other particles, and this may account for some of the challenges in forming a stable dispersion of this material.

### 5.3. Cell Culture Experiments

**Stability in cell culture medium.** The stability of nanoparticles in cell culture medium is critically important to understanding their potential cytotoxic risks. Measuring the size of a nanoform in water or other pure solvent may give a significantly different hydrodynamic radius than is observed in cell culture medium in the presence of salt and serum proteins. Moreover, agglomerates that may form during drying when preparing samples for TEM may not accurately reflect the state of the materials in the biological system. Components of the medium interact with the nanoparticle surface, creating a corona of adherent material at the surface [31]. These interactions can cause particles to agglomerate, cause agglomerated particles to redisperse, or facilitate the dissolution of the particles by removing surface atoms. The rates of these processes are dependent on the composition of the serum (for example, which species it comes from) and its concentration [32,33]. Higher serum concentrations have a greater capacity to chelate free metal ions and may reduce cytotoxicity from dissolved ions. Serum concentrations may also alter the ionic strength of the medium which can impact the stability of the particles towards agglomeration.

From the TEM images in Figure 1, we see that the particles are not uniformly sized or shaped with significant agglomeration, which may result in challenges in obtaining reliable measurements by DLS; however, it should be noted that a significant degree of agglomeration observed by TEM is due to the deposition of the nanoparticles on the TEM grid and that reasonable sizes and dispersion were achieved for most of the samples in water (Table 1). All of the NiO particles that we measured agglomerated quickly in the cell culture medium (less than 30 min), resulting in a layer of precipitated particles at the bottom of the cuvette used for DLS. DLS measurements (Appendix A) were made immediately upon mixing the water suspension of particles with the appropriate amount of cell culture medium so as to mimic the conditions used for the in vitro assays. Subsequent measurements were made on the same samples after 24, 48, and 72 h incubations; however, the particles were resuspended by pipetting the suspension up and down several times to redisperse the precipitated agglomerates. Of the eight samples, only three initially measured an average hydrodynamic radius of less than 1 micron—Ni-03, Ni-07, and Ni-08, and even for these the PDI was very large. This may indicate that even over the ten-minute time course of the measurement, significant sedimentation was occurring. Ni-03 and Ni-08 are bare nanoparticles while Ni-07 is a silane-coated material. There was no obvious difference from the TEM images (Figure 1) between these three samples and the other samples that agglomerated more quickly to account for this different behaviour. As shown above (Table 1), of the eight total samples, six showed reasonable dispersion in water. Of those six, Ni-03 and Ni-07 had the largest hydrodynamic diameter while Ni-08 had the smallest. Within 24 h, both Ni-03 and Ni-07 increased in hydrodynamic diameter to over one micron, possibly indicating that these initially more heavily agglomerated materials in water agglomerated more slowly in medium compared to better dispersed materials. The Ni-08 dispersion, the best dispersed sample in water, was able to maintain a submicron size for 3 days, although the high PDI values prevented a proper quantitative assessment. While we have previously used DLS measurements in cell culture medium to measure changes in silver nanoparticle dispersions, these heavily agglomerated NiO particles were not well-suited for this approach [32,33].

**Cytotoxicity.** Two cytotoxicity assays were performed on two different cell lines. The assays were selected to measure different cellular endpoints that are related to cell viability. The MTT assay measures mitochondrial enzymatic activity while the neutral red assay measures the viability of a cell by its ability to uptake and retain the lysosomal staining dye. We also attempted to use a lactose dehydrogenase assay that measures membrane integrity; however, we were not able to compensate for interferences from the nanoparticles and obtained very inconsistent results for this assay. A549 cells are often used as a model cell line for lung exposure, a common route of entry for particles into the body, while J774A.1 macrophages were used because they are known to rapidly take up nanoparticles and inform on the effect that nanoparticles may have on the immune system. A549 cells have been found to be both sensitive and insensitive to NiO particles in recent reports [13,14,15]. Performing the MTT assay on A549 cells (Table 3, Figure 3), we found that the NiO nanoparticles are indeed toxic at concentrations typically greater than 50 µg/mL. There was a significant increase in the measured cytotoxicity between 24 and 48 h, with no samples causing a 50% reduction in cell viability at 24 h. This suggests that studies that only report 24 h exposures may fail to capture the full extent of the impact of the particles on the cells. Between 48 and 72 h, very little difference was observed. The two samples that exhibited the highest cytotoxicity in this cell line were Ni-01 and Ni-03. Interestingly, these two samples are very different in their physical properties. Ni-01 is reported to be 50 nm in diameter; however, our TEM measurements suggest the diameter is actually less than 20 nm. This sample was also very poorly dispersed in water and cell culture medium and contained a significant fraction of metal impurities. By contrast, Ni-03 had nearly identical size by TEM but was much better dispersed in water and agglomerated more slowly in cell culture medium. The least cytotoxic sample was the stearic acid-coated sample, not surprising considering its very poor dispersibility in water or cell culture medium. Comparing the results from the MTT assay to the size measurements by DLS and TEM, there was no correlation between particle size and cytotoxicity. Indeed, by TEM there was no significant change at all in primary particle size. As for stability, three of the four most cytotoxic samples were the three samples that had hydrodynamic diameters less than a micron by DLS, suggesting some correlation to the agglomerate stability in cell culture medium, though the Ni-01 sample was an outlier and exhibited significant cytotoxicity despite being poorly dispersed.

We also performed a neutral red assay for each material for 72 h (Table 3). This assay measures cell membrane integrity. Here, the results showed less overall cytotoxicity, with all samples showing greater than 50% cell viability at 250 µg/mL, the highest concentration tested. In fact, only Ni-01 showed less than 80% cell viability at that concentration. This suggests that while mitochondrial activity may be impaired, the cells appear to still be intact. This highlights the need to use multiple cytotoxicity assays when performing assessments of nanoparticles as different assays measure different markers linked to cytotoxicity. Ionic controls were performed with nickel sulphate for both the MTT and neutral red assays. The IC_50_ value under our experimental conditions for NiSO_4_ was 110 ± 10 µM at 24 h and 85 ± 10 µM at 48 and 72 h. This increase in cytotoxicity between 24 and 48 h was consistent with some of the NiO samples and could suggest that free ions are the ultimate form leading to cytotoxicity; however, it has been recently shown that the amount of NiO that dissolves in cell culture medium is very low [34]. In order to determine if particle dissolution occurs inside or outside the cell, a detailed nanoparticle uptake study is needed to determine how quickly the nanoparticles are taken up compared to their rate of dissolution in cell culture medium. Unlike for particles, the ionic control gave the same IC_50_ values in the neutral red assay as the MTT assay, suggesting that the mechanism, again, is particle dependent and not just a result of the particles dissolving in cell culture resulting in toxic-free ion concentrations.

The J774A.1 macrophage cell line showed a higher sensitivity to NiO nanoparticle exposure (Table 4, Figure 3) compared to the A549 cells under identical conditions. Here, the IC_50_ values for four of the nanoparticles were lower—Ni-03, Ni-04, Ni-05, and Ni-07, and activity was observed already at 24 h for these samples. Ni-01 again showed significant cytotoxicity, but cell viability did not fall below 50% until the 48 h time point. Three samples showed less cytotoxicity in this cell line. Stearic acid-coated Ni-06 was very poorly dispersed in aqueous media. The other two coated samples, Ni-05 and Ni-07, showed similar cytotoxicity to the uncoated material of the same size from the same manufacturer, suggesting that the coating is not important for assessing the cytotoxicity in this cell line unless it prevents the particle from dispersing as the stearic acid coating does. Ni-02 also showed very poor dispersibility in water despite being uncoated. Ni-08 stood out against any trends as it was reasonably well-dispersed in cell culture medium and water, had approximately the same particle size as the other uncoated samples, yet exhibited less cytotoxicity. One reason for this may be that the nanoparticle agglomerates are driving the cytotoxicity, and that this sample does not agglomerate as much as other more toxic samples and thus the cells are less affected by it. Agglomerates are often excluded from consideration when considering the bioavailability of nanoparticles; however, since macrophages move across the surface of the wells, they may directly contact agglomerates of particles and endocytose them, resulting in a large and rapid increase in internalized Ni concentration. Ni-02 and Ni-06 may be too heavily agglomerated to interact correctly with the cells, excluding them from being more cytotoxic, and leaving those particles with an intermediate level of agglomeration to have the highest toxic effects—large enough to settle to the surface, yet small enough to still interface with the cellular pathways for uptake. The IC_50_ from the ionic controls in this cell line were the same as in A549 cells. Differences in the cytotoxicity of the nanoparticles between the cell lines shows that the nanoparticles themselves must be responsible for the observed cytotoxicity as opposed to dissolution in cell culture medium, as the rate of extracellular dissolution should be the same in both systems.

**Oxidative Stress.** Just as not all of the NiO nanoparticles shared the same cytotoxic behaviour, the measurement of ROS also varied. In the A549 cells, there are reports that indicate that NiO nanoparticles both do and do not result in oxidative stress [13,14,15]. These results can be harmonized, at least somewhat, by our data as, typically, results are reported only for a single nanoform and in some cases, only at a single concentration. In our experiments, we measured the ROS production of eight nanoforms across seven concentrations (Appendix A). Interestingly, there is not a direct link between the amount of ROS produced and the cytotoxicity of the particles in this cell line. This means that the particles are likely interacting with cellular molecules and either directly or indirectly inhibiting critical processes, resulting in cell death. While Ni-01 produced significant ROS and had high cytotoxicity, Ni-03 was even more cytotoxic but produced only minimal ROS compared to untreated controls and only at the very highest concentration tested. Ni-06, the stearic acid-coated sample, did not exhibit any significant cytotoxicity over 72 h; however, this sample still produced ROS in a concentration dependent manner. When we compare the ROS data against the physical chemistry data, we see that Ni-01, Ni-02, and Ni-06 produced higher levels of oxidative stress and were poorly dispersed in water and cell culture medium, while the other samples showed little effect in ROS production outside of the 250 µg/mL samples. Ni-02 is the same sample (different lot) from Sigma Aldrich that has been previously described as not causing oxidative stress [14], although in this report only 20 ug/mL was tested. In our experiments, we observed that the effect was minimal at that concentration, but as the concentration increases, significant effects were seen for this material that are absent in other NiO samples. By acquiring such large data sets for different nanoforms, we can now start to harmonize some of the literature that appears to conflict but often includes too few concentrations or nanoforms or uses different cell lines which makes realistic comparisons impossible. Our results also emphasize the importance of good physical characterization of materials in order to properly assess their biological effects and the need for larger data sets to establish trends in size and coating effects in different cell lines.

In J774A.1 cells, the effect is more pronounced, with much more oxidative stress at higher particle concentrations (Appendix A). Here, again, while more samples showed oxidative stress, there was not an obvious correlation between oxidative stress and cytotoxicity. In fact, for samples such as Ni-04, Ni-05 and Ni-07 that exhibited significant cytotoxicity, while some ROS increase was observed at higher particle concentrations, no measurable effect was observed at 20 µg/mL, the concentration where half the cells were dead (Table 4). This shows that while sometimes an effect is observed, there is a disconnect between the concentrations where ROS is observed and where cell viability is dropping. An ionic control using NiS0_4_ also showed no significant ROS production at concentrations that were cytotoxic to both cell lines. NiO nanoparticles have been previously shown to not dissolve in cell culture medium [15] but to dissolve rapidly under lysosomal conditions, suggesting that particles are likely releasing ions inside the cells after uptake, resulting in a process that is cytotoxic but that does not cause oxidative stress. Ni-02 and Ni-06 also exhibited lower cytotoxicity compared to the other samples, while they also exhibited very high levels of ROS production in this cell line. When comparing the ROS data to particle agglomeration, we again see no evidence for a trend. Ni-03, Ni-04, Ni-07, and Ni-08 exhibited the lowest levels of ROS production, and with the exception of Ni-04, the other three samples had Z-average smaller than one micron initially in cell culture medium. Interestingly, samples Ni-03, Ni-04, and Ni-08 are also all uncoated samples. Ni-07 is coated, but the amine-functionalized particles behaved in a manner more like the uncoated samples, while the PVP- and stearic acid-coated samples showed significant ROS production yet very different cytotoxicities in the MTT and neutral red assays. While Ni-01 and Ni-02 are also uncoated, they stood out as being poorly dispersed in water. This suggests that bare particles produce less oxidative stress if well-dispersed in water, and that the silane-coated sample, Ni-07, behaves similar to a bare, well-dispersed particle, while stearic acid and PVP change the behaviour of the particles in cell culture medium.

## 6. Conclusions

This study explored the toxicity of eight commercially produced NiO nanoparticles. TEM analysis revealed that, despite reported differences from the manufacturers, the primary particles for all nanoforms were approximately the same size. DLS measurements identified clear differences in the ability to form stable dispersions of the powders in water, and even the best dispersed samples showed significant agglomeration. The samples became even more heavily agglomerated immediately upon dilution into cell culture medium. NiO nanoparticles exhibited significantly higher levels of cytotoxicity in J774A.1 cells compared to A549 cells, as well as higher levels of ROS production; however, there was not an obvious relationship between cytotoxicity and ROS production in either cell line. There was a correlation between ROS production and the stability of bare particle dispersions in cell culture medium, and particle agglomerates appeared to contribute to the measured cytotoxicity of adherent macrophage cells. This manuscript highlights physical differences between nanoforms, their behaviour in suspension, and their impact on cellular biology. While several previous studies have asserted clear relationships between specific physical properties and cytotoxicity or ROS production, we have shown that upon expanding the data set to several nanoforms, such correlations no longer hold, and that the cytotoxicity of NiO nanoparticles is not directly linked to particle size, charge, or surface coating, but may instead require a more detailed understanding of particle agglomeration and how this impacts particle uptake and cellular interactions.

## Figures and Tables

**Figure 1 nanomaterials-12-01822-f001:**
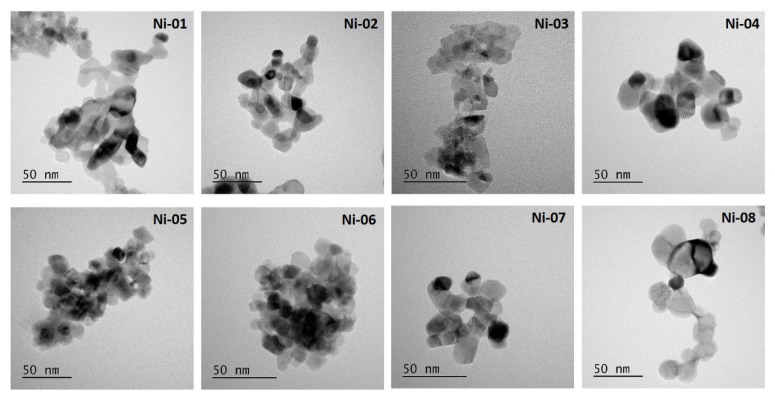
Representative TEM micrographs of NiO nanoparticles dispersed by sonication in water and deposited on TEM grids.

**Figure 2 nanomaterials-12-01822-f002:**
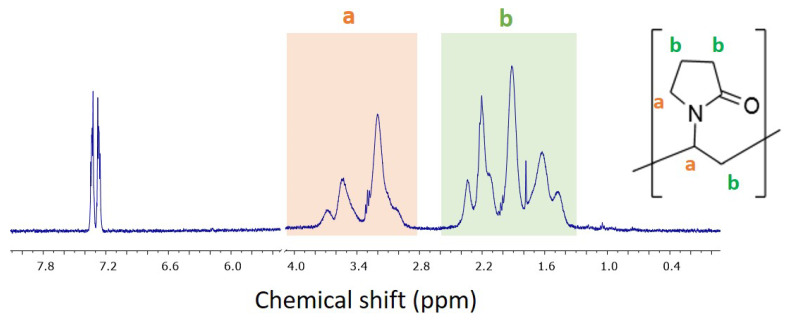
Representative ^1^H NMR spectrum of PVP removed from the nanoparticle surface for Ni-05; note that the region between 4.1 and 5.6 ppm was removed for clarity and the signal due to internal standard being at 7.3 ppm. a and b denote different regions of the proton NMR spectrum and the positions of the polymer to which they correspond.

**Figure 3 nanomaterials-12-01822-f003:**
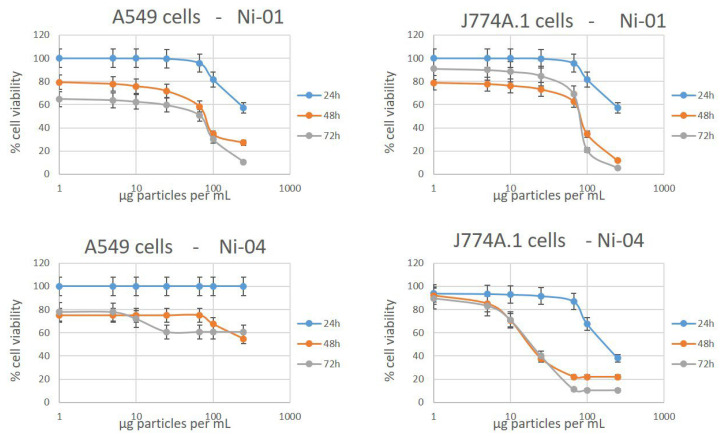
MTT toxicity data from Ni-01 and Ni-04 in both cell lines to show the contrast in toxicity profiles between both samples and cell lines that results in changes in their respective IC_50_ values.

**Table 1 nanomaterials-12-01822-t001:** Sample description and the summary of data provided by the supplier and characterisation performed in house as described in experimental section.

Code	Supplier and Coating	Sonication Energy (J/mL)	Expected Size, Supplier (nm)	TEM ^1^		Dynamic Light Scattering		Zeta Potential (mV)	pH	SSA, Supplier (m^2^/g)	SSA, BET (m^2^/g)
				Equivalent Circular Diameter (nm)	Standard Deviation	Z-Average (nm)	PDI				
Ni-01	Skyspring Nanomaterials	524	50	15.1	5.2	~400	>0.5	22.9 ± 0.7	6.2	~20	24.5
Ni-02	Sigma-Aldrich	525	90	18.5	7.7	~500	>0.5	34.4 ± 0.5	6.2	-	16.1
Ni-03	mkNano	350	<50	12.0	4.3	328 ± 16	0.37 ± 0.04	34 ± 2	6.3	-	77.2
Ni-04	USRN	665	18	20.5	9.2	234 ± 3	0.27 ± 0.02	27 ± 1	6.4	50–100	65.4
Ni-05	USRN—PVP	610	18	16.8	11.8	275 ± 2	0.27 ± 0.02	21.4 ± 0.6	6.6	50–100	57.8
Ni-06	USRN—Stearic Acid	595	18	13.3	6.3	273 ± 14	0.36 ± 0.03	13 ± 1	-	50–100	3.4, 0.6
Ni-07	USRN—Amine	635	18	21.0	13.6	384 ± 10	0.36 ± 0.04	30.4 ± 0.7	6.4	50–100	32.3
Ni-08	USRN	525	15–35	18.8	5.9	158 ± 4	0.32 ± 0.04	34.2 ± 0.8	6.4	50–100	38.7

^1^ TEM size distributions are reported as the average equivalent circular diameter, with standard deviation as a measure of the distribution width. The number of particles (n) analyzed varied from 100 to 160 for all samples except for Ni-03, for which n = 40.

**Table 2 nanomaterials-12-01822-t002:** Surface coating determination via quantitative solution NMR.

Sample and Coating	NMR, µmol/g (n)	Coating Content (Molecules/nm^2^) ^1^
Ni-05, PVP	880 ± 60 (n = 2)	10.6
Ni-06, Stearic acid	457 ± 20 (n = 3)	5.5
Ni-07, Silane	633 ± 33 (n = 2)	7.6

^1^ The values in molecules/nm^2^ were obtained using the NMR data, the estimated surface area from the TEM equivalent diameter, and the density provided by the manufacturer.

**Table 3 nanomaterials-12-01822-t003:** IC_50_ values for NiO samples (µg/mL) in A549 cells from MTT and neutral red assays. Values were determined by plotting cell viability versus concentration at each time point for each sample using an untreated control as 100% and determining the concentration at which 50% of the cells were viable.

	Ni-01	Ni-02	Ni-03	Ni-04	Ni-05	Ni-06	Ni-07	Ni-08
MTT assay at 24 h	>250	>250	>250	>250	>250	>250	>250	>250
MTT assay at 48 h	70 ± 10	240 ± 20	40 ± 10	>250	230 ± 20	>250	110 ± 10	150 ± 20
MTT assay at 72 h	60 ± 10	200 ± 20	50 ± 10	>250	240 ± 20	>250	130 ± 10	140 ± 20
Neutral red assay at 72 h	>250	>250	>250	>250	>250	>250	>250	>250

**Table 4 nanomaterials-12-01822-t004:** IC_50_ values for NiO samples (µg/mL) in J774A.1 cells from MTT and neutral red assays. Values were determined by plotting cell viability versus concentration at each time point for each sample using an untreated control as 100% and determining the concentration at which 50% of the cells are viable.

	Ni-01	Ni-02	Ni-03	Ni-04	Ni-05	Ni-06	Ni-07	Ni-08
MTT assay at 24 h	-	-	100 ± 20	180 ± 20	110 ± 20	-	140 ± 20	-
MTT assay at 48 h	80 ± 10	190 ± 20	30 ± 10	20 ± 10	20 ± 10	150 ± 20	20 ± 10	150 ± 20
MTT assay at 72 h	80 ± 10	190 ± 20	30 ± 10	20 ± 10	20 ± 10	150 ± 20	20 ± 10	160 ± 20
Neutral red assay at 72 h	100 ± 20	-	50 ± 20	50 ± 20	20 ± 10	165 ± 20	40 ± 10	150 ± 20

## Data Availability

Data is available from the corresponding author upon request.

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
