# Peer review of "Physical Characterization and Cellular Toxicity Studies of Commercial NiO Nanoparticles"

_nanomaterials, 2022, doi:10.3390/nano12111822_

Round 1

Reviewer 1 Report

This manuscript reports eight kinds of NiO nanoparticles from different company and with different coatings. Their corresponding physical properties, stability properties in cell culture medium, cytotoxicities and oxidative stresses are characterized. The data is plenty and the results have practical significance. I suggest this manuscript to be accepted before major revision.

  1. Some publications have reported that NiO may may cause cancer. The authors should give explaination that they choose NiO to investigate the corresponding cytotoxicities.
  2. The representative TEM images should be provided in the main text.
  3. The corresponding cytotoxicities and oxidative stresses should be attributed to the different coatings. The authors should give in-depth explaination on the mechanism.
  4. MTT toxicity experiments may be affected by many factors. Thus, more other experiments with other methods should be given to prove the inclusion.
  5. From the TEM images we may see that the purchased NiO nanoparticles are of irregular distributon, uneven size and easy agglomeration. The author should distinguish for the readers about the significance of the investigation on the stability in cell culture medium.

Author Response

We thank the reviewer for their comments and suggestions, please find our responses below.

This manuscript reports eight kinds of NiO nanoparticles from different company and with different coatings. Their corresponding physical properties, stability properties in cell culture medium, cytotoxicities and oxidative stresses are characterized. The data is plenty and the results have practical significance. I suggest this manuscript to be accepted before major revision.

  1. Some publications have reported that NiO may cause cancer. The authors should give explanation that they choose NiO to investigate the corresponding cytotoxicities.

We agree and have added both text and references to the introduction introducing the role of NiO in contributing to certain cancers and causing inflammation in lung cells in animal studies.

  1. The representative TEM images should be provided in the main text.

We agree and have moved the high resolution images to the main text while leaving the lower resolution in the supporting information as an additional reference point for readers.

  1. The corresponding cytotoxicities and oxidative stresses should be attributed to the different coatings. The authors should give in-depth explanation on the mechanism.

We have added additional text to the discussion highlighting the differences between surface coatings and bare particles.  In order to truly elucidate mechanisms for these materials, we believe that more surface coatings would need to be studied, but some potential trends are now more clearly described. Added text:  "Stearic acid-coated Ni-06 is very poorly dispersed in aqueous media.  The other two coated samples Ni-05 and Ni-07 show similar cytotoxicity to the uncoated material of the same size from the same manufacturer suggesting that the coating is not important for assessing the cytotoxicity in this cell line unless it prevents the particle from dispersing as the stearic acid coating does.  Ni-02 also showed very poor dispersability in water despite being uncoated. "

  1. MTT toxicity experiments may be affected by many factors. Thus, more other experiments with other methods should be given to prove the inclusion.

We have also included data from a neutral red assay and highlighted that more.  We have added discussion about other assays that were tried.  We also stress that the MTT assay is commonly used in other reports, and it is important to include to make more direct comparisons to other published literature. Added text: "The assays were selected to measure different cellular endpoints that are related to cell viability.  The MTT assay measures mitochondrial enzymatic activity while the neutral red assay measures the viability of a cell by its ability to uptake and retain the lysosomal staining dye.  We also attempted to use a lactose dehydrogenase assay that measures membrane integrity; however, we were not able to compensate for interferences from the nanoparticles and obtained very inconsistent results for this assay. "

  1. From the TEM images we may see that the purchased NiO nanoparticles are of irregular distribution, uneven size and easy agglomeration. The author should distinguish for the readers about the significance of the investigation on the stability in cell culture medium

We agree and have added/modified text in the section discussing the cell culture medium stability studies:  "From the TEM images in Figure 1, we see that the particles are not uniformly sized or shaped with significant agglomeration, which may result in challenges to obtain reliable measurements by DLS; however, it should be noted that a significant degree of agglomeration observed by TEM is due to the deposition of the nanoparticles on the TEM grid and that reasonable sizes and dispersion were achieved for most of the samples in water (Table 1)."

Reviewer 2 Report

The aim of the manuscript "Physical characterization and cellular toxicity studies of commercial NiO nanoparticles" was to investigate NiO nanoparticles from different manufacturers with different sizes and surface coatings to assess their cellular toxicity as well as the improved properties of three coated NiO particles (coated with polyvinylpyrrolidone, stearic acid and silane). Such an approach provides an assessment of how the particles might behave and can identify potential risks posed by modified coated particles as well as unmodified ones. The authors have succinctly explained the motivation and results of their study.

However, this work also highlights the limitations of current analytical techniques for these types of systems. There are a few questions and points I would like to ask and suggest:
Namely, the authors used different techniques to characterise the coating, such as TEM, DLS, BET. However, additional information from X-ray photoelectron spectroscopic measurements would be of great interest for the studied system, i.e., the authors should perform XPS measurements to verify the chemistry of the deposited layer for coating characterization. Also, the stability of the modified NiO particles can be studied by electrochemical impedance spectroscopy (EIS).

Author Response

We thanks the reviewer for their thoughtful comments and suggestions:

The aim of the manuscript "Physical characterization and cellular toxicity studies of commercial NiO nanoparticles" was to investigate NiO nanoparticles from different manufacturers with different sizes and surface coatings to assess their cellular toxicity as well as the improved properties of three coated NiO particles (coated with polyvinylpyrrolidone, stearic acid and silane). Such an approach provides an assessment of how the particles might behave and can identify potential risks posed by modified coated particles as well as unmodified ones. The authors have succinctly explained the motivation and results of their study.

However, this work also highlights the limitations of current analytical techniques for these types of systems. There are a few questions and points I would like to ask and suggest:
Namely, the authors used different techniques to characterise the coating, such as TEM, DLS, BET. However, additional information from X-ray photoelectron spectroscopic measurements would be of great interest for the studied system, i.e., the authors should perform XPS measurements to verify the chemistry of the deposited layer for coating characterization. Also, the stability of the modified NiO particles can be studied by electrochemical impedance spectroscopy (EIS).

We thank the reviewer for their suggestion to use XPS.  Indeed we have tried a number of analytical techniques.  In our experience we found that the combination of quantitative NMR along with thermogravimetric analysis is a reliable way for determining both the chemical composition and quantity of surface coatings.  We have added information referring to the fact that XPS nitrogen content agrees well with qNMR for the aminopropyl-silane NPs in some of our previously published work but that stearic acid could not be quantified due to hydroxyl content (which makes the O signal complicated) and minor surface contaminants that make the carbon region problematic.  References are included to support these findings.  We have added data from ICPMS analysis of the particles to give complimentary information identifying impurities in the bulk particles, not just at the surface. 

We also are thankful for the reviewers pointing out the possible use of EIS for measuring particle stability.  We do not currently have access to a system for performing such measurements, however we will investigate this further for future projects. For now we rely on DLS for making these assessments, or UV-Vis when the particles remain in suspension without settling (not the case with NiO).  We have also considered TEM to try and make these assessments, however, drying of TEM samples can result in changes to particle agglomeration making these results difficult interpret.

Round 2

Reviewer 1 Report

Accept as it is.

Reviewer 2 Report

Accept in present form